# Genomic Characteristics of Stx2e-Producing *Escherichia coli* Strains Derived from Humans, Animals, and Meats

**DOI:** 10.3390/pathogens10121551

**Published:** 2021-11-28

**Authors:** Xi Yang, Yannong Wu, Qian Liu, Hui Sun, Ming Luo, Yanwen Xiong, Andreas Matussek, Bin Hu, Xiangning Bai

**Affiliations:** 1State Key Laboratory of Infectious Disease Prevention and Control, National Institute for Communicable Disease Control and Prevention, Chinese Center for Disease Control and Prevention, Beijing 102206, China; yangxicdc163@163.com (X.Y.); qianliu9608@gmail.com (Q.L.); sunhui@icdc.cn (H.S.); xiongyanwen@icdc.cn (Y.X.); 2Yulin Center for Disease Control and Prevention, Yulin 537000, China; 2839980@163.com (Y.W.); lm13877560799@163.com (M.L.); 3Division of Laboratory, Medicine Institute of Clinical Medicine, University of Oslo, 0372 Oslo, Norway; anmatu@ous-hf.no; 4Division of Laboratory Medicine, Oslo University Hospital, 0372 Oslo, Norway; 5Shandong Center for Disease Control and Prevention, Jinan 250014, China

**Keywords:** Shiga toxin, *Escherichia coli*, Stx2e, whole genome sequencing

## Abstract

Shiga toxin (Stx) can be classified into two types, Stx1 and Stx2, and different subtypes. Stx2e is a subtype commonly causing porcine edema disease and rarely reported in humans. The purpose of this study was to analyze the prevalence and genetic characteristics of Stx2e-producing *Escherichia coli* (Stx2e-STEC) strains from humans compared to strains from animals and meats in China. Stx2e-STEC strains were screened from our STEC collection, and whole-genome sequencing was performed to characterize their genetic features. Our study showed a wide distribution of Stx2e-STEC among diverse hosts and a higher proportion of Stx2e-STEC among human STEC strains in China. Three human Stx2e-STEC isolates belonged to O100:H30, Onovel26:H30, and O8:H9 serotypes and varied in genetic features. Human Stx2e-STECs phylogenetically clustered with animal- and food-derived strains. Stx2e-STEC strains from animals and meat showed multidrug resistance, while human strains were only resistant to azithromycin and tetracycline. Of note, a high proportion (55.9%) of Stx2e-STEC strains, including one human strain, carried the heat-stable and heat-labile enterotoxin-encoding genes *st* and *lt*, exhibiting a STEC/enterotoxigenic *E. coli* (ETEC) hybrid pathotype. Given that no distinct genetic feature was found in Stx2e-STEC strains from different sources, animal- and food-derived strains may pose the risk of causing human disease.

## 1. Introduction

Shiga toxin-producing *Escherichia coli* (STEC) is a significant foodborne pathogen that can cause disease in humans and animals. The infection of STEC in humans may result in uncomplicated, bloody or non-bloody diarrhea, and even severe fatal sequelae, such as the hemolytic uremic syndrome (HUS) [1]. More than 400 STEC serotypes have been reported globally, of which O157 remains the most common serogroup in clinical cases; however, non-O157 STEC infection has been increasingly reported in recent years [2,3]. The key virulence factor of STEC is Shiga toxin (Stx), which plays a central role in STEC-associated illness. Stx is encoded in the lysogenized lambdoid prophages late region. The Stx-converting prophages, as horizontal gene transfer (HGT) elements, can convert a harmless commensal into an enteric pathogen and promote the emergence of hybrid pathotypes. Thus Stx-converting prophages are considered key drivers of STEC pathogenesis [4]. 

Stx can be categorized into two immunologically distinct types: Stx1 and Stx2. Stx1 and Stx2 can be further classified into various subtypes, four Stx1 subtypes (Stx1a, Stx1c, Stx1d, and Stx1e) and nine Stx2 subtypes (Stx2a to Stx2i and Stx2k) [5,6,7,8]. Different Stx1/Stx2 subtypes vary in toxin potency and host specificity [5]. Stx2 is more often associated with human illness compared with Stx1. Stx1a, Stx2a, Stx2c, and Stx2d are significantly associated with severe clinical outcomes such as hemorrhagic colitis (HC) and HUS, whereas other subtypes are linked to mild symptoms [9,10]. Stx is an AB_5_ toxin. The B pentamer binds to cell surface-exposed glycosphingolipids (GSLs) of the globoseries and is responsible for delivering the A subunit to the cytoplasm [11]. By targeting microvascular endothelial cells of the human kidneys and the brain after gastrointestinal infection, Stx can cause renal insufficiency that can culminate into HUS, with a possible fatal outcome [12]. Of note, Stx1 and most Stx2 subtypes recognize the Stx receptor globotriaosylceramide (Gb3); however, Stx2e binds preferentially to globotetraosylceramide (Gb4), and can specifically recognize the uncommon Stx receptor globopentaosylceramide (Gb5) [9,12]. The distinctive difference in binding affinity confers Stx2e a unique recognition feature among other Stx subtypes. Stx2e is the main subtype associated with porcine edema disease [13]. Strains producing Stx2e have been widely reported in swine and pork in many countries [13,14]. Stx2e-STECs have also been found in environmental samples such as wastewater [15]. Although Stx2e-STEC is less frequently identified in humans, a few studies have indicated an association between Stx2e-STEC and human HUS [16], mild diarrhea, or asymptomatic cases [17,18,19], but the relationship between STEC isolates and human diseases has not been clearly delineated.

Human STEC infections may occur through direct contact with animals and consumption of animal-derived foods, especially raw meat. In a previous study [20], 6.2% of healthy pigs carried STEC strains, and all pig-derived STEC strains carried the *stx*_2e_ subtype. A prior study indicated that Stx2e-producing *E. coli* isolates from humans and pigs differ in their virulence profiles [21], which may partially explain the varied pathogenicity among different hosts. However, the molecular characteristics of Stx2e-STEC from diverse sources are not fully understood. The objectives of the present study were to analyze the prevalence of Stx2e-STEC strains in humans and other sources in China and to characterize the molecular features of human-derived Stx2e-STEC strains in comparison with animal- and food-derived strains using whole-genome sequencing (WGS). The phylogenetic relationships of Stx2e-STEC strains from different hosts were assessed, and their antimicrobial resistance was examined.

## 2. Results

### 2.1. Prevalence of stx_2e_-Carrying STEC Strains in Diverse Hosts 

In total, 176 STEC strains carrying *stx*_2e_ were identified from a collection of 818 STEC strains (21.5%) (Table 1). Stx2e-STEC strains were isolated from different samples collected from eight provinces in China between 2012 and 2018. Of note, 98% and 91.7% of STEC strains from pig and pork, respectively, carried the *stx*_2e_ subtype. Stx2e-STEC strains were also isolated from cattle (2.9%), goat (1%), mutton (14%), and beef (18.5%). Three out of 44 (6.8%) human-derived STEC strains in our collection carried *stx*_2e_, two were from diarrheal outpatients hospitalized in Shanghai, and one was from a healthy carrier in Shenzhen (Figure 1).

Three Stx2e-STEC strains (one from a diarrheal patient, one from beef, and one from goat) also belonged to the *stx*_2k_ subtype. Two strains (both from pig) carried two identical *stx*_2e_, and four strains carried two different *stx*_2e_ variants each. A 1331 bp insertion sequence (IS*2* family transposase) was found in the intergenic regions between the *stx*_2_ A and B subunits in three strains (all from raw meat).

### 2.2. Human-Derived Stx2e-STEC Strains

The human-derived Stx2e-STEC strains belonged to different serotypes (O100:H30, Onovel26:H30, and O8:H9) and MLST types (ST993, ST710, and ST21) and varied in their genome structures. The chromosome size of the three strains was ~4.9 Mbp. A total of 4753 (strain STEC409), 4826 (strain STEC413), and 5003 (strain STEC509) protein-encoding genes were predicted. The genomes comprised 20, 17, and 8 prophages, respectively. The plasmids, rRNA, and tRNA of the three human-derived Stx2e-STEC strains are summarized in Table 2.

### 2.3. Molecular Characteristics of Stx2e-STEC Strains from Different Sources

Twenty-three different serotypes were found in 59 Stx2e-STEC strains, among which, O9:H30 was the most predominant serotype accounting for 15% (9/59) of Stx2e-STEC strains, followed by O8:H19 (6/59, 10%), O100:H30, O9:H4, and Onovel26:H30 (5/59, 8%). The same serotypes were found in human-derived Stx2e-STECs (e.g., O100:H30 and Onovel26:H30 from diarrheal patients, O8:H19 from a healthy carrier) and strains from animals and meat (Figure 1). Twenty-two MLST sequence types (STs) were assigned to 59 Stx2e-STEC strains. ST933 (11/59, 19%) was the most predominant sequence type, followed by ST710 (8/59, 14%). Two strains from diarrheal patients (STEC409 and STEC413) belonged to ST933 and ST710; no other strain shared the same sequence type (ST21) with strain STEC509 from a healthy carrier (Figure 1).

Multiple virulence factors genes were identified in the 59 Stx2e-STEC strains. These virulence genes were classified into several groups based on their functions: adherence, iron uptake, secretion system, toxin, invasion, and others (Appendix A). In addition to Stx2, other toxin-encoding genes were identified, for example, a gene cluster encoding alpha-hemolysin, *astA* (heat-stable enterotoxin 1, EAST1), *sta* (heat-stable enterotoxin STa), *stb* (heat-stable enterotoxin STb), *lt* (heat-labile enterotoxin LT). The adherence-associated factors mainly included porcine attaching and effacing associated factors Paa, curli fibers, type 1 fimbriae, P fimbriae, CFA/I fimbriae, etc. Secretion system effectors mainly included type II secretion proteins, non-LEE-encoded type III secretion system effectors, and type VI secretion system factors (Figure 1 and Appendix A).

Strikingly, thirty-three Stx2e-STEC strains carried heat-labile toxin (LT)- or/and heat-stable toxin (ST)-encoding genes, which are virulence determinants for ETEC, thereby exhibiting a hybrid STEC/ETEC pathotype. Four STEC/ETEC strains (three from meat and one from an animal) carried the heat-labile toxin-encoding gene *lt-II*, 28 strains (one from a patient, 6 from meat, and 21 from animals) carried the heat-stable toxin gene *sta*, and 9 strains (five from animals and four from meat) carried the heat-stable toxin gene *stb.* Eight strains (five from animals and three from meat) carried both *sta* and *stb.*

### 2.4. Phylogenetic Relationships of the Stx2e-STEC Strains from Different Hosts

To assess the phylogenetic relationships of the Stx2e-STEC strains from humans and other hosts reported in this study and elsewhere, whole-genome SNP-based phylogeny trees were constructed. All Stx2e-STEC genomes available in the NCBI database were downloaded and used for comparison. The analysis showed that the Stx2e-STEC strains from different sources in this study were inter-mixed, yet, strains from foods or animals exhibited a higher tendency to cluster closely. The three human-derived strains distributed separately; among these, strain STEC413 from a diarrheal patient clustered with animal-derived strains, four of them exhibited the same serotype (Onovel26:H30) and ST (ST710) with STEC413, a total of 833 SNPs were found between strain STEC413 and a pig-derived strain STEC383. Strain STEC409 clustered with one strain from animal and three strains from meat, and 1670 SNPs were found between strain STEC409 and strain STEC351 from beef. One strain from the healthy carrier (STEC509) formed a separate cluster. The majority of Stx2e-STEC genomes were grouped based on serotype, with a few exceptions (Figure 1). In line with the phylogenetic relationships of Stx2e-STEC strains in this study, Stx2e-STEC strains from other countries scattered throughout the phylogenetic tree, and no distinct cluster was observed based on the host of strains, with a few exceptions (Appendix A). 

### 2.5. Genetic Feature of Stx2e-Converting Prophages

We obtained 30 complete sequences of Stx2-coverting prophages from 22 complete Stx2e-STEC genome sequences. Of these, 28 Stx2e prophages and two Stx2k prophages were predicted, which was consistent with the *stx* subtyping showing that two strains (one from a diarrheal patient and one from beef) carried both *stx*_2e_ and *stx*_2k_ subtypes, and six strains harbored two copies of the *stx*_2e_ gene. The 28 Stx2e-converting prophages were further characterized in terms of chromosomal insertion site, genetic structure, and sequence diversity. The size of 28 Stx2e prophages ranged from 24,820 bp to 83,659 bp, and the predicted CDSs ranged from 54 to 131. Five different insertion sites were found in the 28 Stx2e prophages. Three Stx2e prophages were integrated into the coding sequence of the *parB* gene (ParB/RepB/Spo0J family partition protein), 5 were integrated into the *potC* gene (spermidine/putrescine import ABC transporter permease protein PotC), 3 were integrated into the *yccA* gene (Modulator of FtsH protease YccA), 13 were inserted in the *yciD* gene (outer membrane protein W), and 4 were inserted in the *yecE* gene (DUF72 domain-containing protein YecE). For the six strains harboring two copies of the *stx*_2e_ gene, the two Stx2e prophages in each strain were genetically diverse, with, e.g., different insertion site, length, etc. (Appendix A).

The Stx2e prophages from three human strains were genetically diverse. We then selected Stx2e prophages from other sources in this study that shared similar molecular features and close phylogenic relationship with the human-derived Stx2e prophages, together with two reference Stx2e prophages P27 (from a diarrheal patient) [22] and S1191 (from a pig with edema disease) [23]. The comparison between Stx2e prophages from humans and other sources revealed a considerable diversity of Stx2e prophages among strains sharing similar genomic features. Each prophage could be divided into three general modules, including integration and regulatory genes, virulence and lytic genes, DNA-packaging and morphogenesis, as previously reported [24]. The genetic structures of regulatory and virulence regions were similar among different Stx2e prophages, while regions related to morphogenesis were variable (Figure 2).

### 2.6. Antimicrobial Resistance of Stx2e-STEC Strains

Among the 19 antibiotics tested in this study, the resistance rate toward tetracycline was the highest (76.2%, 44/59), followed by resistance to nalidixic acid (59.3%, 35/59) and trimethoprim–sulfamethoxazole (47.5%, 27/59). The resistance rate to chloramphenicol, ampicillin, colistin, and azithromycin was 20.3% (12/59), 20.3% (12/59), 16.9% (10/59), and 15.3% (8/59), respectively. All isolates were susceptible to aztreonam, ciprofloxacin, ertapenem, meropenem, amikacin, ampicillin–sulbactam, cefoxitin, ceftazidime, ceftazidime-avibactam, and imipenem. Thirteen isolates (22.0%) were susceptible to all 19 antimicrobial agents tested. Five isolates (8.5%) were only resistant to one antimicrobial substance. One diarrheal patient-derived Stx2e-STEC strain was resistant to azithromycin and tetracycline, the healthy carrier-derived strain was resistant to tetracycline, and one diarrheal patient-derived strain was susceptible to all antibiotics tested. Multidrug resistance was identified in 35 non-human isolates (59.3%) (Table 3). 

The antibiotic resistance phenotype corresponded to the presence of antibiotic resistance-related genes for some antibiotics. For instance, strains carrying genes associated with resistance to phenicols (*cmlA1*, *cmlA6*, and *floR*), tetracyclines (*tetA* and *tetD*), macrolides (*mphA* and *mrx*), were all resistant to chloramphenicol, tetracyclines, and azithromycin, respectively. Eight out of nine isolates carrying colistin-related genes (*mcr-1* and *mcr-3.1*) were resistant to colistin. However, the majority of strains carrying AMR genes involved in resistance to class C β-lactamase (*ampC*), aminoglycoside (*aadA*, *aac(3)-IIa* and *aac(3)-IIc*) were not resistant to related antibiotics (Table 3).

## 3. Discussion

Stx2e-producing *E. coli* strains have been identified from various sources including animals [25,26,27], foods [28,29,30,31], and environment, especially wastewater [32]. Human infections with Stx2e-STEC strains are rare. It has been reported that Stx2e-STEC isolates accounted for 4.6% (12/262) of STEC strains isolated from patients with diarrhea and 4.2% (4/96) of STEC strains from asymptomatic individuals in Germany [17]. Bai et al. recently reported that 1.1% (2/184) of clinical STEC strains carried *stx*_2e_ in Sweden [33]. In this study, 6.8% of total human-derived STEC strains carried *stx*_2e_, which is slightly higher than the percentage reported in other countries, as mentioned above. This might be due to the higher transmission of Stx2e-STEC strains through more frequent contact with animals or consumption of contaminated meat in China. The swine breeding industry is thriving in China, and numerous *E. coli* isolates from pigs with edema disease and post-weaning diarrhea have been recovered, with Stx2e-STEC accounting for a large proportion of them [34,35]. Moreover, high rates of Stx2e-STEC strains were reported from healthy pigs and raw meat in China [20,36]. These may contribute to the higher number of human Stx2e-STEC infections in China. However, we acknowledge that only three human sourced isolates were used in this study, thus the related data should be compared with caution.

Although Stx2e is a less common subtype in human STEC strains, studies have shown associations between Stx2e-STEC and severe symptoms such as HUS [16,37,38], and diarrhea [18,19]. Molecular characterization revealed that human-derived Stx2e-STEC strains in this study were diverse. The serotypes of human-derived Stx2e-STEC strains were also identified in strains from other sources. It has been found that a vast majority of Stx2e-STEC isolates do not express virulence factors associated with HUS, such as genes encoding intimin (*eae*) and enterohemorrhagic *E. coli* hemolysin (*ehxA*) [39]. It is therefore likely that Stx2e-STEC cause disease through different mechanisms [40]. Consistent with previous studies [39], *eae* and *ehxA*, commonly detected in O157:H7 strains, were absent in all Stx2e-STEC strains in this study. All our Stx2e-STEC strains belonged to non-O157 serotypes. No difference was observed between human and pig isolates in the iron uptake system-encoding genes *fyuA* and *irp1*/*irp2*, dissimilar to an earlier study [21]. Further studies are required to investigate the pathogenic mechanisms of Stx2e-STEC.

The whole-genome phylogeny indicated a high diversity of Stx2e-STEC strains, with strains from different hosts scattered throughout the phylogenetic tree. The human-derived strains clustered with animal- and food-derived strains, and no host-specific cluster was observed. A similar trend was found when comparing our Stx2e-STEC strains with strains from different sources in other countries. We were unable to clearly distinguish Stx2e-STEC strains associated with human disease and strains from other sources at genomic level in this study. It should be noted that one O100:H30 strain from a diarrheal patient in our study clustered with O100:H30 strains isolated from diarrheal patients in Sweden and Netherlands [41,42]. Besides, some animal and food-derived strains in this study were phylogenetically close to human-derived Stx2e-STEC strains from other countries. Serotype-specific clusters were observed among some serotypes, e.g., O121:H10, O139:H1, and O155:H21, while some were not clustered, e.g., O8:H19, O9:H10, and O100:H30, indicating the genetic diversity of these Stx2e-STEC strains.

Notably, 56% of Stx2e-STEC strains in our study, including one of three human strains, carried the heat-stable toxin (ST) or/and heat-labile toxin (LT)-encoding gene *st* or/and *lt*, exhibiting a STEC/ETEC hybrid pathotype. It should be noted that Stx2 and most ST/LT encoding genes are located on phages and plasmids, where horizontal gene transfer may occur, thereby facilitating the emergence of hybrid pathotypes. On the other hand, the high prevalence of a hybrid pathotype among Stx2e-STEC strains may indicate a greater plasticity of Stx2e-STEC genomes compared to those of other subtypes. Further studies are warranted to explore the mechanism of the emergence of hybrid pathotypes and the high prevalence of STEC/ETEC hybrids among Stx2e-STEC strains.

Stx phages are known to have similar morphologies (e.g., short or long non-contract tails). Their genomes size ranged from 30 to 70 kb and showed little homology, although having a similar genetic organization [43]. We observed genetic diversity mainly in the phage morphogenesis region among Stx2e phages from different hosts, but also from the same host. Five different insertion sites were found in 28 Stx2e phages, four of which have been reported in Stx phages previously [44]. Consistent with the report that a significant portion of STEC strains carry more than one Stx phage [24], we found that eight strains harbored two Stx2 prophages. Intriguingly, two Stx2e-STEC strains harbored a prophage carrying the recently identified *stx*_2k_ subtype [7]. This expands the prior knowledge that *stx*_2e_ was rarely coexistent with other *stx* genes [39]. IS elements were identified in the *stx* gene in three Stx2e prophages. The 1.3 kb transposable element IS*2* is a member of the IS3 family [45]. Pinto et al. [24] reported that the IS3 family is widely distributed in Stx phages. Further studies are required to examine Stx2 production in Stx2e-STEC strains and to assess if the expression of Stx2e could be inhibited by insertion elements. 

In this study, the antibiotic resistance phenotype corresponded well to the genotype for some antibiotics, such as colistin, chloramphenicol, tetracycline, and azithromycin, while strains carrying genes involved in resistance to beta-lactamase antibiotics, aminoglycosides, and sulfonamides were largely susceptible to the related antimicrobial agents. This may be due to the inhibited expression level of genes, as previously reported [46,47]. Cointe et al. [48] reported that azithromycin decreases Stx production at subinhibitory concentrations, suggesting that it could be tested in clinical trials. Yet, one diarrheal patient-derived Stx2e-STEC strain in our study was resistant to azithromycin, suggesting caution in the use of azithromycin in the treatment of STEC-infected patients suffering from diarrhea and HUS. It should be noted that 59.3% of Stx2e-STEC strains showed multidrug resistance (MDR), and all MDR strains were isolated from animals and foods. This might be due to a higher antibiotic exposure in animals. The high prevalence of MDR isolates highlights the importance of a proper management of antibiotics use in farming. The animal- and food-derived MDR Stx2e-STEC strains may pose a high risk of causing severe disease.

To our knowledge, this is the first study reporting the prevalence and molecular traits of Stx2e-STEC strains from diverse hosts. Our study demonstrated a wide distribution of Stx2e-STEC in diverse hosts. Genomic characterization revealed considerable genetic diversity of Stx2e-STEC strains from different sources, and human-derived Stx2e-STEC strains clustered with animal- and food-derived strains. Of note, a high prevalence of STEC/ETEC hybrid pathotypes was found among the Stx2e-STEC strains, indicating great plasticity of the Stx2e-STEC genomes. Given that no distinct genetic feature was found in the human Stx2e-STEC strains, human infection with Stx2e-STEC through contact with animals and consumption of contaminated foods should be controlled to attenuate the risk of disease.

## 4. Materials and Methods

### 4.1. Stx2e-STEC Strains Collection

Fecal samples from diarrheal patients, healthy carriers, animals, and meats were collected from April 2009 to March 2019 in different regions of China. STEC strains were isolated and confirmed by the methods as previously described [49]. Briefly, the samples were enriched in EC broth and then tested for the presence of *stx*_1_/*stx*_2_ genes by PCR. *stx*-positive samples were inoculated into two selective media, i.e., CHROMagar^TM^ ECC agar and CHROMagar^TM^ STEC agar (CHROMagar, Paris, France), for isolation of STEC strains, as described previously [42]. After overnight incubation at 37 ℃, presumptive colonies were picked and tested for *stx* genes by single a colony duplex PCR assay. API 20E biochemical test strips (bioMérieux, Lyon, France) were used for a confirmatory test. The *stx*_1_/*stx*_2_ subtyping was initially conducted by amplifying, sequencing the complete *stx*_1_/*stx*_2_ genes, and comparing against known *stx* subtypes, as described previously [5]. STEC strains carrying the *stx*_2e_ subtype were selected for subsequent study.

### 4.2. Whole-Genome Sequencing (WGS), Assembly, and Annotation 

Bacterial DNA extraction was performed with the Wizard Genomic DNA purification kit (Promega, Madison, WI, USA) according to the manufacturer’s protocol. Library preparation and WGS were performed at Beijing Novogene Bioinformatics Technology Co., Ltd., China. To obtain the draft genomes of STEC, DNA library preparation was done using NEBNext® Ultra™ DNA Library Prep Kit (New England Biolabs, Ipswich, MA, USA). The library was then pair-end (2 × 150 bp) sequenced using the Illumina NovaSeq 6000 platform (Illumina, San Diego, CA, USA). The paired-end reads were filtered by fastp 0.20.1 (https://github.com/OpenGene/fastp accessed on 25 January 2021) [50] and then *de novo* assembled using SKESA version 2.4.0 (https://github.com/ncbi/SKESA accessed on 25 January 2021) [51]. Low-quality contigs (length < 500 bp) were filtered with Seqkit version 0.11.0 [52]. To obtain the complete genomes, two sequencing libraries were prepared, in addition to the Illumina DNA library as described above, a 10 kb library was prepared using an SMRT bell Template Prep kit (version 1.0), and then sequenced using the PacBio Sequel platform (Pacific Biosciences, Menlo Park, CA, USA). The long reads were preliminarily filtered using “RUN QC” module in SMRT Link version 5.1.0 (www.pacb.com/support/software-downloads accessed on 30 January 2021), de novo assembled using the Hierarchical Genome Assembly Process (HGAP) pipeline [53], then corrected using the Illumina short reads to obtain complete genomes. The genome sequences were annotated with Prokka (version 1.11) [54].

### 4.3. Determination of stx_2_ Subtype, Serotype, MLST, Virulence Factor Genes and Antimicrobial Resistance Genes

The *stx* subtypes were further verified by comparing all the assemblies against an in-house database including 149 representative nucleotide sequences of all identified *stx*_1_ subtypes (*stx*_1a_, *stx*_1c_, *stx*_1d_, and *stx*_1e_) and *stx*_2_ (*stx*_2a_ to *stx*_2k_) subtypes using ABRicate version 0.8.10 (https://github.com/tseemann/abricate accessed on 2 May 2021). In silico serotyping and screening of virulence factors and antimicrobial resistance (AMR) genes were conducted using ABRicate version 0.8.10 against EcOH database [55], the *E. coli* virulence factors repository (https://github.com/phac-nml/ecoli_vf accessed on 2 May 2021), and the Comprehensive Antibiotic Resistance Database [56] (http://arpcard.mcmaster.ca accessed on 2 May 2021), respectively, with the following parameters: coverage≥ 90% and identity≥ 80%. The enterotoxin-encoding genes were extracted and compared against an in-house database including representative nucleotide sequences of the heat-stable and heat-labile enterotoxin-encoding gene *st* and *lt*, using ABRicate version 0.8.10. Nucleotide sequences of different *st* and *lt* subtypes were obtained from previous reports [57,58,59]. Multilocus sequence typing (MLST) of seven housekeeping genes was performed through an on-line tool provided by the Warwick *E. coli* MLST scheme website (https://enterobase.warwick.ac.uk/species/ecoli/allele_st_search accessed on 2 May 2021). 

### 4.4. Single-Nucleotide Polymorphism (SNP)-Based Phylogeny 

A whole-genome SNP phylogeny was used to assess the genomic diversity and relatedness of Stx2e-STEC strains in this study. First, the core alignment of the SNPs was obtained by using snippy-multi in Snippy version 4.3.6 (https://github.com/tseemann/snippy accessed on 23 August 2021) with default parameters; the reference genome was Sakai (NC_002695.2). Gubbins version 2.3.4 [60] was then used to remove recombinations from core SNP alignments. Finally, a maximum likelihood tree based on filtered SNP alignments was constructed using FastTree version 2.1.10. (http://www.microbesonline.org/fasttree/ accessed on 23 August 2021). Snippy version 4.3.6 (https://github.com/tseemann/snippy accessed on 2 November 2021) was used for SNP variants calling between human-derived strains and strains from other sources. To assess the phylogenetic position of Stx2e-STEC strains in this study among others, 102 Stx2e-STEC genome sequences from different sources were download from the National Center for Biotechnology Information database (NCBI) (Appendix A).

### 4.5. Genomic Characterization of Stx2e-Converting Prophages

The genome characteristics of Stx prophages were analyzed and visualized using methods previously described [7]. Briefly, PHAge Search Tool Enhanced Release (PHASTER, http://phaster.ca/ accessed on 15 May 2021) was used to identify Stx-converting phages from STEC genomes. The intact Stx prophage sequences were then extracted from the complete genomes. Subsequently, the RAST server (http://rast.nmpdr.org/ accessed on 18 May 2021) [61] was used to annotate the genome of Stx-converting phage. The functions of protein-coding sequences (CDSs) were verified using BLASTP (https://blast.ncbi.nlm.nih.gov/Blast.cgi accessed on 20 May 2021) against the GenBank non-redundant protein database. The gene adjacent to the integrase gene was designated as the phage insertion site [44]. The full sequences of Stx2e phage were compared in detail using Mauve [62] and blastN program in BLAST, with an e-value cutoff of e-10 and was visualized using an in-house Perl script (https://github.com/dupengcheng/BlastViewer accessed on 13 November 2021). 

### 4.6. Antimicrobial Resistance Testing

The BD Phoenix^TM^ M50 Automated Microbiology System (BD, San Jose, CA, USA) was used to determine the minimal inhibitory concentrations (MICs) of 19 antimicrobial agents according to the manufacturer’s instructions, as previously described [63]. The antimicrobial agents include ampicillin–sulbactam (SAM, 1–32 µg/mL),trimethoprim–sulfamethoxazole (SXT, 0.5–8 µg/mL), meropenem (MEM, 0.125–8 µg/mL), colistin (PB, 0.25–8 µg/mL), ceftazidime–avibactam (CZA, 0.25/4–8/4 µg/mL), nitrofurantoin (F, 32–256 µg/mL), tetracycline (TET, 1–16 µg/mL), ertapenem (ETP, 0.25–8 µg/mL), ceftazidime (CAZ, 0.25–16 µg/mL), chloramphenicol (CHL, 4–32 µg/mL), imipenem (IPM, 0.25–8 µg/mL), ampicillin (AMP, 2–32 µg/mL), cefoxitin (FOX, 2–64 µg/mL), cefotaxime (CTX, 0.25–16 µg/mL), nalidixic acid (NA, 4–32 µg/mL), azithromycin (AZM, 2–64 µg/mL), ciprofloxacin (CIP, 0.015–2 µg/mL), aztreonam (ATM, 0.25–16 µg/mL), and amikacin (AMK, 4–64 µg/mL). Multidrug resistance (MDR) was defined as acquired non-susceptibility to at least one agent in three or more antimicrobial categories [64]. 

## Figures and Tables

**Figure 1 pathogens-10-01551-f001:**
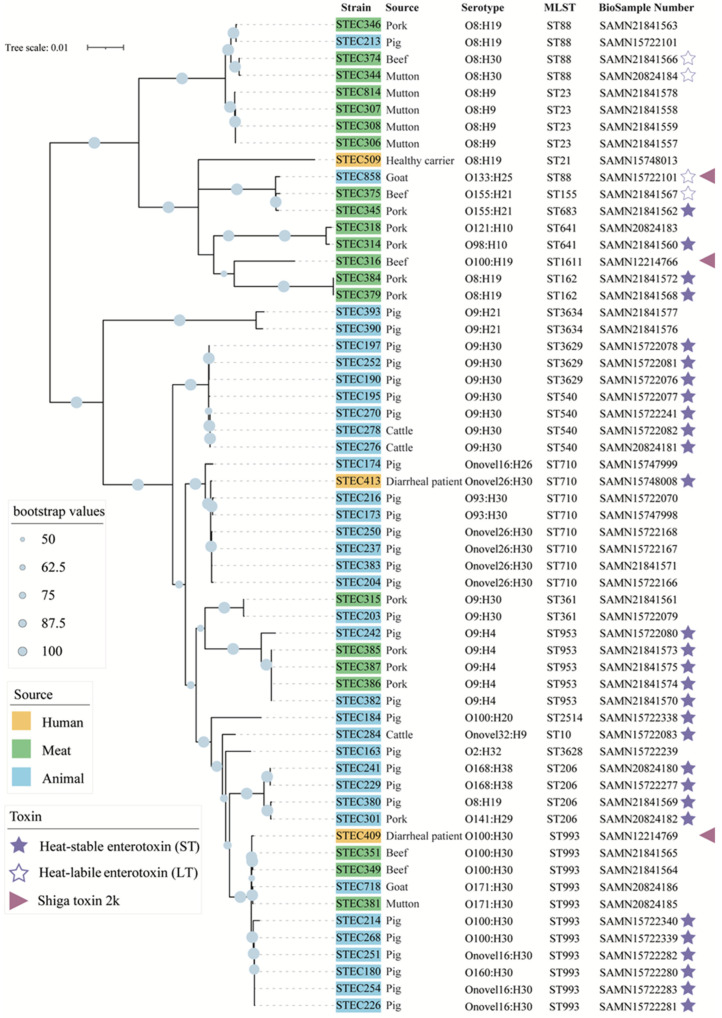
Phylogenetic tree based on core-genome single-nucleotide polymorphisms (SNPs) using the Maximum-Likelihood method. Source, serotype, MLST type, accession number of all strains are shown. Strains carrying heat-stable/heat-labile enterotoxin-encoding genes and the Shiga toxin 2k gene are marked, as indicated. The blue circle on the branch indicates the bootstrap value (≥50%) of the node.

**Figure 2 pathogens-10-01551-f002:**
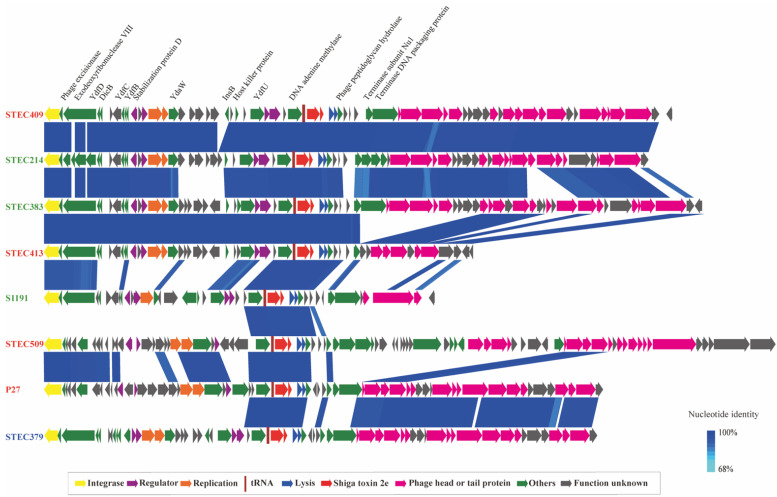
Architecture of eight Stx2e-converting prophages from different hosts. Corresponding CDSs are colored as indicated. The color of the text indicates the source of the strains: red represents human-derived strains, green represents animal-derived strains, and blue represents meat-derived strains. Stx2e prophages P27 (from a diarrheal patient) and S1191 (from a pig with edema disease) were used as references. Stx2e prophages from the three human-derived strains were compared with those from other sources that share similar genetic features and close phylogenic relationship, i.e., strain STEC409 from a diarrheal patient and strain STEC214 from pig share the same serotype (O100:H30) and MLST type (ST993); strain STEC413 (diarrheal patient) and strain STEC383 (pig) share the same serotype (Onovel26:H30) and MLST type (ST710); strain STEC509 (healthy carrier) and strain STEC379 (meat) belong to the same serotype (O8:H19). Ranges of over 68% nucleotide identity between phages are marked by blue shading.

**Table 1 pathogens-10-01551-t001:** Distribution of Stx2e-STEC strains from different hosts in China.

Source		No. of STECs	No. of Stx2e-STECs (%)	No. of Stx2e-STECs Used for WGS ^a^
Human	Healthy carrier	5	1 (20.0%)	1
	Diarrheal patient	39	2 (5.1%)	2
Animal	Pig	147	144 (98.0%)	29
	Cattle	172	5 (2.9%)	3
	Goat	202	2 (1.0%)	2
	Chicken	4	0	0
	Tibetan antelope	5	0	0
	Yak	126	0	0
	Marmot	9	0	0
	Pika	22	0	0
Meat	Pork	12	11 (91.7%)	11
	Mutton	43	6 (14.0%)	6
	Beef	27	5 (18.5%)	5
	Chicken	1	0	0
	Duck	1	0	0
Environment	Water	3	0	0
Total		818	176 (21.5%)	59

^a^ WGS: whole-genome sequencing.

**Table 2 pathogens-10-01551-t002:** Genome features of human-derived Stx2e-STEC strains in this study.

Strain	Total Length (bp)	G + C ^a^ Ratio (%)	No. CDS ^b^	No. rRNA	No. tRNA	No. Prophages	No. Plasmids
STEC409	4,857,389	51.82	4753	22	94	20	2
STEC413	4,948,664	50.61	4826	22	90	17	1
STEC509	4,871,415	50.71	5003	22	87	8	0

^a^ Guanine and cytosine. ^b^ Coding sequence.

**Table 3 pathogens-10-01551-t003:** Prevalence of antimicrobial resistance phenotype and genotype among the 59 Stx2e-STEC isolates.

Antimicrobial Classes	AMR Phenotype	AMR Genes	No. of Phenotypic Resistant Isolates		No. of Phenotypic Susceptible Isolates	
			Resistant by Genotype	Susceptible by Genotype	Resistant by Genotype	Susceptible by Genotype
Tetracyclines	Tetracycline	*tet(A), tet(D)*	6	38	0	15
Quinolones	Nalidixic_Acid	*oqxA, oqxB, qnrD1, qnrS1*	13	22	6	18
Trimethoprim	Trimethoprim-Sulfamethoxazole	*dfrA12, dfrA14, dfrA15, dfrA17*	10	1	19	29
Phenicols	Chloramphenicol	*cmlA1_1, cmlA6, floR*	10	2	0	47
Colistin	Colistin	*mcr-1, mcr-3.1*	8	2	1	48
Macrolides	Azithromycin	*mphA, mrx*	4	4	0	51
Nitrofurans	Nitrofurantoin	*-* ^b^	0	1	0	49
β-lactamase	Ampicillin	*ampC, bla* _ACC-1c_ *, bla* _CTX-M-14_ *, bla* _CTX-M-65_ *, bla* _TEM-116_ *, bla* _TEM-1B_	12	0	41	6
	Cefotaxime					
Aminoglycoside	Amikacin	*aac(3)-IIa, aac(3)-IIc, aadA, aadA2, aadA5, aph(3′)-Ia, aph(3″)-Ib, aph(3′)-IIa, aph(6)-Id*	0	0	32	27
Fosfomycins	- ^a^	*fosA3*	0	0	1	58
Sulfonamides	- ^a^	*sul1, sul2, sul3*	0	0	31	28

^a^ Antimicrobial resistance testing did not include the corresponding antibiotics. ^b^ No corresponding antimicrobial resistance gene was detected.

## Data Availability

The data presented in this study are publicly available in the NCBI, with accession numbers provided in Figure 1.

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
