# Peer review of "Genomic Characteristics of Stx2e-Producing Escherichia coli Strains Derived from Humans, Animals, and Meats"

_pathogens, 2021, doi:10.3390/pathogens10121551_

Round 1
Reviewer 1 Report
Many readers would probably appreciate it if MLST groups were used instead of serotypes, or in addition to serotypes, in Fig 1 and Fig S1. Serotypes are often not as phylogenetically consistent as MLST groups.
For Fig 2, indicate the stx2e subunit genes - I assume they are indicated by the red arrows, but these are labeled as "virulence factor".
The format for Table 4 is challenging in several places, with no borders around cells. On the right side, the headings need more space separating them, and below that several rows have awkward positioning as you go across, which makes it unclear which antibiotic resistance some of the genes are intended to go with. (I could figure them out pretty easily, but a table shouldn't be set up to be confusing...)
Line 247 - The authors need to be specific when they say "the iron uptake system encoding gene". There are many iron uptake systems that have been associated with pathogenesis in E. coli (as shown in Table S2). Based on the reference cited, I think the authors are referring to irp and fyu, which are involved in synthesis of yersiniabactin. This needs to be made clear.
The authors provide a very extensive table (S2) looking at the presence of several hundred virulence factors in 59 sequenced genomes, but there is no significant analysis of this data. The only genes they devote any time to discussing (other than stx2e) are the LT and ST genes. There is no discussion of statistical association of virulence factor genes based on source, serotype, MLST group, etc... This doesn't neatively impact the main point of the paper, but it seems to me a missed opportunity to extract even more value from the genomes reported on herein.
Reviewer 2 Report
The illustrations are complex, extensive and contain the necessary information, yet they are clearly presented. I have no further comments on the manuscript itself.
However, the authors mentioned that one diarrheal patient-derived Stx2e-STEC strain was resistant to azithromycin and tetracycline. The literature (Cointe et al.2020. J Antimicrob Chemother. 2020 Mar 1;75(3):537-542. doi: 10.1093/jac/dkz484) mentions that azithromycin decreases Stx production at subinhibitory concentrations. Cointe at al. (2020) suggest testing it in clinical trials. Given that you identified one Stx2e-STEC clinical isolate resistant to azithromycin, it would be interesting to discuss in few words what possible consequences this may have in the treatment of patients suffering from diarrhea and HUS. In that way, it would be appropriate to add the reference of Cointe et al.2020 in the manuscript when you will discuss this specific point.
Reviewer 3 Report
Summary:
The manuscript titled: “Genomic characteristics of Stx2e-producing Escherichia coli strains from humans compared with strains from animals and meats” by Yang et al., provides a WGS analysis of 59 Stx2e-STECs derived from human (3), animal (34) and meat sources (22) from China. The methods used are all standard procedures that have been previously published, however, the presentations and analyses are fairly superficial. In particular, one of the most interesting observations made was that more than half of the isolates are STEC/ETEC hybrids, however, apart from marking these on a phylogenetic tree, no further detail is provided.
Broad comments:
- I was not able to download the supplementary data files, an error appears stating that the .zip file is empty. I have informed the editors in case this was an issue with the submission process.
- Given that the study only uses 3 human-sourced isolates, the title of the paper should be reconsidered
- Please italicise all gene names (e.g. L128-132)
- Providing some SNP distances between human and non-human sourced isolates would be informative, particularly STEC413 and STEC409, which cluster with non-human sourced isolates of the same serotypes
- It is indeed striking that the study identified 33 STEC/ETEC hybrids and therefore warrants more information, such as what colonising factors (CFs) are associated with what ST/LT subtype. Additionally, as ETEC virulence factors are carried on plasmids, reporting what plasmid replicons and pMLST types are present would also be very useful. These could be added to Figure 1, or presented separately.
Specific comments:
L39: Haemorrhagic colitis is the main cause of bloody diarrhea
L52-55: The paper the authors referenced for these two sentences states that Stx1a and Stx2a subtypes most commonly cause severe pathogenesis in humans.
L54: change “outcome” to “outcomes”
L57: change “delivery of” to “delivering”
L59: change “with possible” to “with a possible”
L66: change “been also” to “also been”, and change “waste water” to “wastewater”
L67: Remove comma after “Although”
L68-70: This sentence is confusing
L73: change “pig” to “pigs”, and remove “of” before “pig-derived”
L85: Is the disease status of the animals from this collection known?
Figure 1: Most of the information in Table 3 (MLST and source) could be added to Figure 1 and all metadata (including Biosample numbers) could be provided in a supplementary data file (thereby removing table 3). Please also include bootstrap values in Figure 1.
L127: Add “the” before “59”
L158-161: Why would this phylogenetic analysis not be included?
Figure 2: Please include the nucleotide similarities in this Figure. I am also assuming that “virulence factor” in red refers to the shiga toxin/toxins? This should be made clear.
Table 4: There appears to be an error in the “No. of strains” column (many "0" values)
L227-229: If the authors want to make this comparison, they should note that the number of human isolates in this study was considerably fewer.
L231-233: This sentence is confusing. Rewrite.
L234: change “rate” to “rates”
L241: change “don’t” to “do not”
L243: change “it’s” to “it is”
L272: change “are ranging” to “ranged”
Round 2
Reviewer 3 Report
The authors have attended to most of my previous comments and the manuscript has been approved. However, some changes should be made before I can recommend acceptance for publication in Pathogens.
I have attached a word document with my additional comments

Round 3
Reviewer 3 Report
The authors have satisfied my previous comments and suggestions.
This manuscript is a resubmission of an earlier submission. The following is a list of the peer review reports and author responses from that submission.